# Applying Affordance Factor Analysis for Smart Home Speakers in Different Age Groups: A Case Study Approach

Chih-Fu Wu [1], Ying-Kit Wong [2,*], Hsiu-Hui Hsu [3] and Cheng-Yu Huang [1]

[1] Department of Industrial Design, Tatung University, Taipei City 104, Taiwan; wcf@gm.ttu.edu.tw (C.-F.W.); g10604010@ms.ttu.edu.tw (C.-Y.H.)
[2] Graduate Institute of Design Science, Tatung University, Taipei City 104, Taiwan
[3] Department of Applied Cosmetology, Lee-Ming Institute of Technology, New Taipei City 243, Taiwan; alice@mail.lit.edu.tw
[*] Correspondence: d10717004@ms.ttu.edu.tw

**Abstract:** Many people use smart speakers at home nowadays for various reasons, such as playing music, checking news and weather, setting timers/alarms, etc. However, before smart speakers were created and available on the market, people used to have home audio systems for similar applications. Nonetheless, the control systems of smart speakers have many different appearances. Affordance is the information given by an object, which is determined by its appearance and supplies clues about its appropriate operation. Therefore, smart speakers should have affordances. Since smart speakers are the main device in the sustainable lifestyle of human beings in smart homes, this study analyzed the affordances of its appearance affect people and the result is essential to the sustainability of smart home. The present study presents a review of the smart speakers in Taiwan, focusing on the four main affordances (physical, cognitive, sensory, functional) and three different age groups (60 participants) based on four appearance categories of smart speaker control, namely, mechanical button control, no-button–no-touch control, touchscreen control, and touch sensor control. By examining the comparison of three age groups, 18–24, 25–49 and 50+, the results of one-way ANOVA showed that the smart speakers with touchscreen control and touch sensor control had a significant difference ($p < 0.01$) in four main affordances among these three age groups. The smart speakers with mechanical button control and no-button–no-touch control had no significant difference ($p > 0.01$) in four main affordances among these three age groups. In conclusion, age-range and cultural group affect the affordance of smart home speakers.

**Keywords:** smart speaker; smart home; home audio; home stereo; affordance; affordance-based design

## 1. Introduction

### 1.1. Smart Speakers and Home Audio Systems

Today, people can use smart speakers to listen to music and enjoy different types of entertainment comfortably at home. Home audio and speaker systems have evolved over many years. In the past few decades, people were more familiar with a general home audio system that can play music on vinyl records, cassette tapes, CDs, etc. This original concept began in 1877, when Thomas Edison invented the phonograph, which was the first machine in the world that could play music. About 10 years later, in 1887, the disc-spinning turntable was invented by Emil Berliner. This was the first gramophone in the world that could play music [1]. In the beginning, the phonograph was very expensive. Cheaper phonographs for home use were introduced in 1894, by Columbia, and in 1896, by Edison [2,3].

Home audio systems, home stereo systems or high-fidelity (Hi-Fi or HiFi) are mostly used to describe home audio equipment, which allows the high-quality reproduction of sound [4]. People can enjoy music or songs at home through such devices.

Nowadays, many users are using smart speakers to replace the traditional home audio system for playing music. There are many other different kinds of functionality in smart

speakers, but playing music is the most popular application for smart speaker users. A study has shown that half of US homes will have a smart speaker device by 2022 [5,6].

The affordances of home stereo systems are similar to and overlap with those of televisions, which include turning on and off, changing channels, connecting, providing sound and light, adjusting sleep settings, adjusting the volume or picture, and selecting menu options [7]. The control method of general home audio systems is mainly via a mechanical button. Regarding the control methods, previous studies mentioned some methods and mathematical solutions to solve the optimal control problems with uncertainties [8,9].

Affordance can help to simplify the interface mapping and functionality of a product's design. For example, a good design for a power switch button for a home stereo shows the labels clearly and provides more operational details regarding the button [10].

A smart speaker is normally a speaker with a voice command function, which is an integrated virtual assistant that can offer hand-control interaction (requiring an action by the human hand or finger to input the instructions) or hands-free activation by the user speaking some "hot words" or commands to control the system [11]. One of the main purposes is to make the system more convenient, faster and comfortable for the users when they control the smart speaker [12]. For the consumers, the interpretation of the design is mainly based on the users' interactions with the product. Consumers normally have no access to a product design, so it is important for a designer to think and design the product well [13].

The popular uses of smart speakers are listening to music and news, as users are replacing radios with smart speakers [14]. Based on the Smart Audio Report 2020, the majority of smart speaker owners (85%) use the system for playing music, and the next most common tasks for which owners use the smart speaker are for checking the weather (74%), and setting a timer/alarm and checking the time (65% and 62%, respectively) [15]. Playing music on smart speakers is the most frequent application, and it is the same application as that of a general home audio system. Therefore, since both products have a similar application at home, and the outlooks for smart speakers and home audio systems are different, it is important to understand the affordance of smart speakers. There is a value co-creation between smart service and users through the smart speaker assistant, and by combining the innovation of smart technologies, it offers users a new way of interaction to reach their goals [16].

### 1.2. Affordance

Affordance is the basic information provided by the product to users about operating, using, and executing the product. The product's structure is determined by the appearance of the objects, and users can understand its affordance and the purpose of its existence when they look at it and use it [17]. Affordance is not just functional meaning and automatic receiving ability; it is also a process combining emotion, cognition, and interaction [18].

Affordance was firstly mentioned by James J. Gibson in 1966 in his book *The Senses Considered as Perceptual Systems*, and refers to what the environment offers the individual [19]. In 1977, in his book *The Ecological Approach to Visual Perception*, Gibson wrote the definition of affordance, stating that "The affordances of the environment are what it offers the animal, what it provides or furnishes, either for good or ill. The verb to afford is found in the dictionary, but the noun affordance is not. I have made it up. I mean by it something that refers to both the environment and the animal in a way that no existing term does. It implies the complementarity of the animal and the environment" [20].

Gibson provided some good examples in his book. For example, he mentioned the physical situation in which if a surface is nearly horizontal, nearly flat (convex or concave), and sufficiently extended, and its substance is rigid, then the surface affords support. Gibson emphasized the relationship between the environment, object, or creature, and the user, stating that there is a direct connection between them [20].

Whether the creature realizes it or not, or if the function is executed or not, as long as the conditions of both parties exist, their complementary relationship will last forever.

Thus, based on Gibson's explanation, (1) affordance is independent and does not depend on our insight and perception, (2) the existence of affordance is related to our actions, and (3) affordance will not change because of our needs and purposes [20].

In 1988, Donald A. Norman published *The Psychology of Everyday Things*, in which he mentioned that "the term affordance refers to the perceived and actual properties of the thing, primarily those fundamental properties that determine just how the thing could possibly be used . . . Affordances provide strong clues to the operations of things. Plates are for pushing. Knobs are for turning. Slots are for inserting things into. Balls are for throwing or bouncing. When affordances are taken advantage of, the user know what to do just by looking: no picture, label, or instruction needed" [21]. In the later stages of the concept of affordance, human–computer interaction (HCI) design and image design were slowly introduced. In 1999, Norman published the article "Affordance, Convention and Design". In this article, he explained that we should not confuse affordances with perceived affordances, and we should not confuse affordances with conventions. Affordances are what an object or product can do, and perceived affordances are what people think it can do. Affordance comprises both the actual and perceived properties. When the actual and perceived properties are combined, an affordance appears. There are three kinds of behavioral constraints as follows: physical, logical, and cultural. Physical constraints are related to real affordance; logical constraints use reasoning to determine the choices, and cultural constraints are conventions shared by a cultural group [22].

After much academic discussion, affordance became more complex without uniformity. In 2003, Hartson sorted out and analyzed the concept of affordance, and clarified the unclear definitions of affordance used by scholars in the past, as he believed that there were many ideas about affordance that needed to be clarified [23] (Table 1).

**Table 1.** Comparison of affordance terminology.

| Name | Physical | Cognitive | Sensory |
|---|---|---|---|
| Hartson [23] | Physical affordance | Cognitive affordance | Sensory affordance |
| Gibson [19,20] | Affordance | Perceptual information about an affordance | Implied |
| Norman [21,22] | Real affordance | Perceived affordance | Implied |
| McGrenere and Ho [24] | Affordance | Perceptual information about an affordance | Indirectly included in the perceptibility of an affordance |
| Gaver [25] | Affordance and perceptible affordance | Perceptual information about an affordance, and apparent affordance | Indirectly included in the perceptibility of an affordance |

Note: Adapted from Hartson (2003) [23].

McGrenere and Ho targeted the current misuse and confusion of terms used in discussing affordance, and compared the fundamental difference between the definitions of affordance given by Gibson and Norman. They found Gibson was interested in how people perceive the environment, whereas Norman was interested in manipulating or designing the environment so that its usefulness can be perceived easily [24].

Gaver divided affordance into perceptible affordance, hidden affordance, and false affordance. He created a framework for separating the affordances from the perceptual information [25].

In addition, a previous study shows the spatially distributed uses of smart speakers were discovered through the users' perception of spatial affordances. Additionally, this is related to the process of externalization for the use of networked devices [26].

Hartson, based on the definitions of affordance given by Gibson (1977), Norman (1988), McGrenere and Ho (2000), and Gaver (1991), organized the information from other

scholars and concluded that there are four types of affordance, namely cognitive affordance, physical affordance, sensory affordance, and functional affordance, as shown in Table 2 [23]. The definition given by Hartson (2003) can be summarized as follows:

1.  Cognitive affordance helps and encourages people to think and understand things. It mainly helps users to use their judgment and understand an object's function. For example, clear words or labels on buttons can help users understand the meaning of the button, and then understand the function of the button or feedback results.
2.  Physical affordance helps and encourages people to actually perform actions and carry out tasks. It can help users to understand the operational role of their skills. For example, the size and the relatively accessible position of a button allow the user to click on the button more easily.
3.  Sensory affordance helps and enhances what can be seen or felt by people. It helps the user to understand the perceptions of their senses, such as sight, touch, hearing, smell, or other sensory abilities.
4.  Functional affordance increases the direction, purpose, and awareness of a product or object, and it helps people to use the product effectively. In order to achieve this goal, the user must sense and understand the affordance. For example, designers use the restriction of fool-proofing a design to prevent misuse or accidents [23].

**Table 2.** Summary of affordance types.

| Affordance Type | Description | Example |
| --- | --- | --- |
| Cognitive affordance | Design features that help users know something | A button label that helps users know what will happen if they click on it |
| Physical affordance | Design features that help users conduct a physical action via the interface | A button that is large enough that users can click on it accurately |
| Sensory affordance | Design features that help users sense something (especially cognitive affordances and physical affordances) | The label's font size large enough to read easily |
| Functional affordance | Design features that help users accomplish a task (i.e., the usefulness of a system's function) | The internal system's ability to sort a series of numbers (invoked by users clicking on the sort button) |

Note: Adapted from Hartson (2003) [23].

### 1.3. Affordance-Based Design Method

Maier and Fadel presented a generalized theory of affordances applicable to design in a series of articles [27–33], and pioneered affordance-based design based on Gibson's and Norman's theories of affordance.

In this systematic design system, the affordance-based design method (ABD) represents the affordance relationships of the designer, user, and artifact (DAU) (Figure 1). Affordance properties link the artifacts and designers, and affordance which is needed links designers and users, alongside AUA (artifact–user affordances) between artifacts and users.

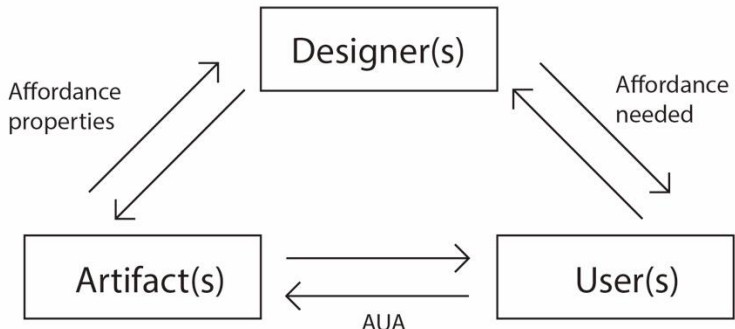

**Figure 1.** Affordance relationships in the designer–artifact–user (DAU) system. Note: adapted from Maier, J. R. (2006) [30].

The affordance-based design method can be used and adopted in product design, because the product design process is also about the designer–product–user relationship [34]. For smart devices used at home, because there is an interaction between the users and the smart home devices, there is affordance in this process [35].

Furthermore, when the designer designs products via the affordance-based design method, usability is also important to consider. Alongside perceptions and ergonomics, ABD can help produce a better product design [36]. Moreover, in product redesign, which is necessary in some special cases, using the affordance mindset in the redesign process is beneficial [37].

*1.4. Technology Adoption in Different Age Groups*

People of different ages make different decisions about the use of technological products [38]. A smart speaker is a technological product, so when users are operating smart speakers, they will make different decisions.

According to a 2019 Pew Research Study, there are five generation groups as follows: the Silent Generation (1928–1945), Baby Boomers (1946–1964), Generation X (1965–1980), Generation Y (1981–1996), and Generation Z (1997–2012) [39]. Generation Z, unlike other generations, live in an era when many technologies are available and easy to access [40]. Generation Y (also called Millennials) are called digital natives, which means they are very familiar with digital technology products [41]. Generation X are described as highly understanding of technology, and proficient in computers and the internet. Generations X and Y both have a higher rate of adoption of mobile technology than Baby Boomers [42,43]. Compared with Generations X, Y, and Z, Baby Boomers spend the least time online on the internet and using mobile multimedia systems [44].

In 2018, PricewaterhouseCoopers (PwC), which is one of the Big Four accounting firms in the global market, hired a leading global research firm to survey 1000 people in the US as a representative sample between the ages of 18 and 64 to investigate consumer adoption and usage of voice assistance technology. The questionnaire asked how often they used voice assistance technology devices, and the research results were divided into three age groups. For the first age group (18–24 years old), the answers indicated 59% heavy usage, 33% medium usage, and 8% light usage. For the second age group (25–49 years old), the answers indicated 65% heavy usage, 29% medium usage, and 6% light usage. For the third age group (50+), the answers indicated 57% heavy usage, 40% medium usage, and 3% light usage (Figure 2) [45]. The heaviest usage of voice assistance technology was by people aged 25–49 years old [45,46].

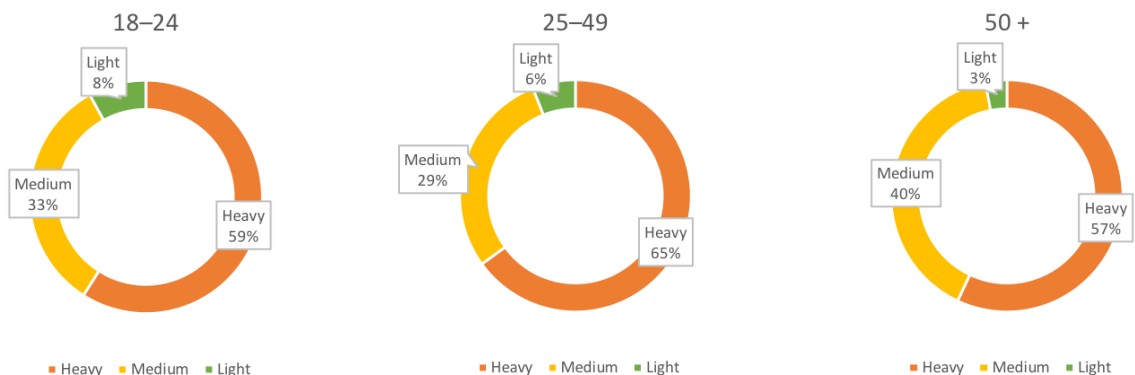

**Figure 2.** The usage rates of three age groups for voice assistance technology. Note: sourced from PwC (2018) [45].

This survey shows that the three age groups have different usage levels, suggesting that it is important to present a comparison of these three age groups.

Most of the investigation of smart speakers are related to the technical analysis and voice control functionality, and there is no other research related to the smart speaker's appearance and analysis of the four main affordances in different age groups in Taiwan. This paper reviews smart speakers in Taiwan, focusing on the four main affordances (physical, cognitive, sensory, functional) and three different age groups, based on four appearance categories of smart speaker control, namely, mechanical button control, no-button– no-touch control, touchscreen control, and touch sensor control.

## 2. Materials and Methods

### 2.1. Sample Collection

We collected information about the smart speakers that are available in Taiwan. The smart speakers are either for sale in stores or online shops in Taiwan. Various details of each sample were documented as follows: (1) name or model number, (2) company or brand, (3) size, (4) photos of the appearance, (5) control method, (6) feedback method, (7) color, and (8) material.

Name or model number was necessary to record, because a company might produce many smart speakers with different model names or different versions. Company or brand indicated the name of the company that made the smart speaker. Size was recorded marked in millimeters in the following 3 different dimensions: depth (D) × width (W) × height (H). The appearance of different perspectives of the speakers was recorded in multi-view photos (isometric view, front view, top view, side view). There are different control methods and visible feedback methods in different smart speakers. Color indicated the outer case color. Material indicated what the outer case was made of.

### 2.2. Professional Experts' Expertise in Smart Home Devices

Virzi (1992) and Nielsen and Landauer (1993) gave the popular advice that five participants are enough for an experiment, allowing 80% of problems to be detected. They mentioned that best results come from testing no more than five users. For a general study of usability, a group size of 3–20 participants is typically valid, with 5–10 participants being a sensible baseline range [47–51]. Thus, at the beginning of experiment, five professional experts with expertise in smart home devices and technological product design were invited to participate in this research. These experts all have more than 10 years' working experience in the industry and come from five different consumer electronic companies in Taiwan. With their professional knowledge and industry background, they analyzed the samples and divided them into different control method categories, then selected a representative sample for each category based on the affordance, visibility, usability and product details.

### *2.3. Participants*

Sixty participants were invited to take part in this research, who were local residents of Taiwan. They were divided into three age groups, with each group including 20 participants. The age group ranges followed the following three age groups in the PwC survey of voice assistance technology devices, undertaken in 2018 [45]:

1.  Group #1: younger age (18–24);
2.  Group #2: middle age (25–49);
3.  Group #3: older age (50+).

### *2.4. Multi-View Smart Speaker Photos*

In the technical graphics and engineering design field, spatial visualization is important. The isometric view is the most informative view of a three-dimensional object, but the front view, top view, and side view of an object also have many advantages in spatial visualization [52]. The appearance data showed a clear view of the smart speakers through multi-view photos (isometric view, front view, top view, and side view), which showed the different perspectives and angles of the speakers. These photos were used in the experiment as described below:

1.  The professional experts looked at the multi-view photos of all the collected samples and then divided them into different control method categories;
2.  The professional experts looked at the multi-view photos of all the collected samples and then selected a representative sample of each control method category;
3.  All 60 participants looked at the multi-view photos of each representative sample in the experiment.

### *2.5. Survey and Questions*

The survey used the affordance questions from the article by Hartson [23], which included 3 specific questions related to 3 affordances (physical, cognitive, and functional affordance) and 2 specific questions related to 1 affordance (sensory affordance). The 2 sensory affordance questions were combined into 1 question for this experiment, and thus there were 4 affordances questions representing each affordance in this survey. During the survey, participants looked at the multi-view photos of the smart speakers when they answered the questions. Each question rated on a 7-point Likert scale.

#### 2.5.1. Physical Affordance Question

The article by Hartson asked whether the artefact was easy to manipulate by all users in the target user classes [23]. We converted the Hartson question into a question for this particular survey to enquire about physical affordance, asking "Is this smart speaker easy for you to manipulate?"

#### 2.5.2. Cognitive Affordance Question

The article by Hartson included the following question: "Does the design include clear, understandable cues about how to use the artefact?" [23]. We adapted this question for our survey question to examine cognitive affordance, asking "Does this smart speaker design include clear, understandable cues about how to use it?"

#### 2.5.3. Sensory Affordance Question

Hartson's article asked whether users can easily sense the visual (or other) cues about an artefact's operation or its manipulation [23]. We rewrote this question for our survey question to enquire about sensory affordance, asking "Can you easily sense the visual cues about this smart speaker's operation or its manipulation?"

### 2.5.4. Functional Affordance Question

Hartson's original article asked whether the functionality to which this interaction or artefact gives access is useful for achieving the user's goals through performing a task [23]. We adapted this question for our survey to explore functional affordance, asking "Is the functionality of this smart speaker useful for achieving your goals (any of the functions of the smart speaker) through performing tasks?"

### 2.6. Statistical Method

Means and ANOVA were used in this experiment to check if there were significant differences among the 3 age groups of users, in terms of the 4 different affordances (physical, cognitive, sensory, and functional affordance), for each representative smart speaker sample.

## 3. Results

### 3.1. Descriptive Statistics

In total, the details and photos of 26 smart home speakers either available in stores or online shops in Taiwan were collected. Table 3 provides (1) the name or model number, (2) company or brand, (3) size, and (4) photos of the appearance.

**Table 3.** Details of the smart speakers: name/model, company/brand, size, and appearance.

| Number | Name/Model | Company/Brand | Size (mm) D × W × H | Isometric View | Front View | Top View | Side/Back View |
|---|---|---|---|---|---|---|---|
| 1 | AliGenie C1 | Alibaba | 66 × 135 × 60 | | | | |
| 2 | Apple Homepod | Apple | 142 × 142 × 172 | | | | |
| 3 | Amazon Echo Second Generation | Amazon | 88 × 88 × 148 | | | | |
| 4 | Google Home | Google | 96 × 96 × 143 | | | | |
| 5 | Google Home Hub | Google | 67 × 178 × 47 | | | | |
| 6 | Google Home Mini | Google | 98 × 98 × 42 | | | | |

**Table 3.** *Cont.*

| Number | Name/Model | Company/Brand | Size (mm) D × W × H | Isometric View | Front View | Top View | Side/Back View |
|---|---|---|---|---|---|---|---|
| 7 | Amazon Echo Show Second Generation | Amazon | 107 × 246 × 175 |  |  |  |  |
| 8 | Sonos One | Sonos | 120 × 120 × 160 |  |  |  |  |
| 9 | Amazon Echo Dot Third Generation | Amazon | 99 × 99 × 43 |  |  |  |  |
| 10 | Google Home Max | Google | 187 × 335 × 150 |  |  |  |  |
| 11 | Amazon Echo PlusSecond Generation | Amazon | 99 × 99 × 147 |  |  |  |  |
| 12 | Amazon Echo Spot | Amazon | 81 × 104 × 97 |  |  |  |  |
| 13 | Xiao Ai Class-mateSecond Generation | Xiaomi | 88 × 88 × 212 |  |  |  |  |
| 14 | Xiaobao | Cheetah Mobile | 120 × 120 × 149 |  |  |  |  |
| 15 | Xiao Ai Jiang | FET | 105 × 105 × 41 |  |  |  |  |
| 16 | ibobby | Chunghwa Telecom | 114 × 114 × 224 |  |  |  |  |

**Table 3.** *Cont.*

| Number | Name/Model | Company/Brand | Size (mm) D × W × H | Isometric View | Front View | Top View | Side/Back View |
|---|---|---|---|---|---|---|---|
| 17 | AliGenie X1 | Alibaba | 83 × 83 × 126 | | | | |
| 18 | AliGenie M1 | Alibaba | 94 × 94 × 63 | | | | |
| 19 | DingDong2 LS | Jing Dong | 98 × 98 × 162 | | | | |
| 20 | DingDong LS-MC1 | Jing Dong | 110 × 110 × 49 | | | | |
| 21 | Ding Dong LS-TOP2 | Jing Dong | 100 × 100 × 48 | | | | |
| 22 | Ding Dong LS-PLAY | Jing Dong | 197 × 129 × 211 | | | | |
| 23 | Xiao Du XDH-01-A1 | Baidu | 89 × 89 × 105 | | | | |
| 24 | Xiao Du Doss | Baidu | 112 × 112 × 145 | | | | |
| 25 | Xiao Du NV5001 | Baidu | 125 × 192 × 192 | | | | |
| 26 | Xiao Du Pro | Baidu | 90 × 90 × 209 | | | | |

After the professional experts analyzed the affordance, visibility, usability, and product details of the 26 smart speakers, all the smart speaker samples were divided into four categories based on the differences in the appearance of the control method, namely, (1) mechanical button control, (2) no-button–no-touch control, (3) touchscreen control, and (4) touch sensor control. All 26 smart speaker samples had the same function, which is the voice control method. Although the voice control software of these 26 smart speakers varies depending on the different companies or brands, the functionality was the same, and there were no differences among them. In addition, voice control cannot be seen. Therefore, this functionality was not suitable and was not used in this experiment.

The other details of each sample are documented in Table 4, including (1) the control method, (2) the feedback method, (3) color, and (4) material.

**Table 4.** Details of the control method, feedback method, color, and material of the smart speakers.

| Number | Mechanical Button Control | No-Button–No-Touch Control | Touchscreen Control | Touch Sensor Control | Feedback | Color | Material |
|---|---|---|---|---|---|---|---|
| 1 | ● | | | | Light | White | Plastic |
| 2 | | | | ● | Light | Black | Plastic and fabric |
| 3 | ● | | | | Light | Black | Plastic and fabric |
| 4 | | ● | | | Light | White and grey | Plastic and fabric |
| 5 | | | ● | | Screen | White and grey | Plastic and fabric |
| 6 | | ● | | | Light | White and grey | Plastic and fabric |
| 7 | | | ● | | Screen | Black | Plastic and fabric |
| 8 | | | | ● | Light | Black | Plastic and metal |
| 9 | ● | | | | Light | White | Plastic and fabric |
| 10 | | ● | | | Light | White and grey | Plastic and fabric |
| 11 | ● | | | | Light | Black and grey | Plastic and fabric |
| 12 | | | ● | | Screen | Black | Plastic |
| 13 | | | | ● | Light | White | Plastic |
| 14 | | | | ● | Light | Black and grey | Plastic and metal |
| 15 | ● | | | | Light | Black and red | Plastic |
| 16 | | | | ● | Light | Grey | Plastic and fabric |
| 17 | | ● | | | Light | Black and grey | Plastic and fabric |
| 18 | ● | | | | Light | White and grey | Plastic and fabric |
| 19 | | ● | | | Light | White and grey | Plastic and fabric |
| 20 | ● | | | | Light | White and grey | Plastic |
| 21 | | | | ● | Light | White | Plastic |
| 22 | | | ● | | Screen | Black | Plastic |
| 23 | | | | ● | Light | White and grey | Plastic and fabric |
| 24 | | ● | | | Light | White and grey | Plastic and fabric |
| 25 | | | ● | | Screen | White and grey | Plastic and fabric |
| 26 | | | | ● | Light | White and grey | Plastic and fabric |

Note: Representative sample numbers corresponding to those in Table 3.

The results show the number of smart speakers for each control method category:

1. Mechanical button control method smart speakers: seven;
2. No-button–no-touch control method smart speakers: six;
3. Touchscreen control method smart speakers: five;
4. Touch sensor control method smart speakers: eight.

Four representative smart speaker samples were selected by the professional experts, one for each control method category, after they had analyzed the affordance, visibility, usability, and product details of all 26 smart speakers (Table 5).

**Table 5.** The four representative samples.

| Representative Sample # | Category | Name/Model | Company/Brand |
|---|---|---|---|
| 3 | Mechanical button control | Amazon Echo Second Generation | Amazon |
| 6 | No-button–no-touch control | Google Home Mini | Google |
| 12 | Touchscreen control | Amazon Echo Spot | Amazon |
| 16 | Touch sensor control | ibobby | Chunghwa Telecom |

Note: #—Representative sample numbers corresponding to those in Table 3.

The following four representative samples, one for each control method category, were selected by the professional experts:

1. Mechanical button control method: Smart speaker No. 3;
2. No-button–no-touch control method: Smart speaker No. 6;
3. Touchscreen control method: Smart speaker No. 12;
4. Touch sensor control method: Smart speaker No. 16.

*3.2. Inferential Statistics*

3.2.1. Mechanical Button Control Method

For the representative sample of the mechanical button control method, a seven-point Likert scale questionnaire was used in the survey, in which each participant answered all questions. The mean score for physical affordance was 4.9, the score for cognitive affordance was 4.7, that for sensory affordance was 5, and the score for functional affordance was 5.05. The total mean of all four affordances was 4.91 (Table 6, Figure 3).

**Table 6.** Mean results for the smart speaker with the mechanical button control method.

| Affordance | Age Group | N | Mean | SD | SE |
|---|---|---|---|---|---|
| Physical | 18–24 | 20 | 4.35 | 1.226 | 0.274 |
| | 25–49 | 20 | 5.45 | 1.276 | 0.285 |
| | 50+ | 20 | 4.9 | 1.447 | 0.324 |
| | Total | 60 | 4.9 | 1.374 | 0.177 |
| Cognitive | 18–24 | 20 | 4.45 | 1.191 | 0.266 |
| | 25–49 | 20 | 4.8 | 1.576 | 0.352 |
| | 50+ | 20 | 4.85 | 1.424 | 0.319 |
| | Total | 60 | 4.7 | 1.394 | 0.18 |
| Sensory | 18–24 | 20 | 4.6 | 1.429 | 0.32 |
| | 25–49 | 20 | 5.45 | 1.432 | 0.32 |
| | 50+ | 20 | 4.95 | 1.432 | 0.32 |
| | Total | 60 | 5 | 1.45 | 0.187 |
| Functional | 18–24 | 20 | 4.9 | 1.483 | 0.332 |
| | 25–49 | 20 | 5.3 | 1.49 | 0.333 |
| | 50+ | 20 | 4.95 | 1.538 | 0.344 |
| | Total | 60 | 5.05 | 1.489 | 0.192 |

Note: N, number of participants; SD, standard deviation; SE, standard error.

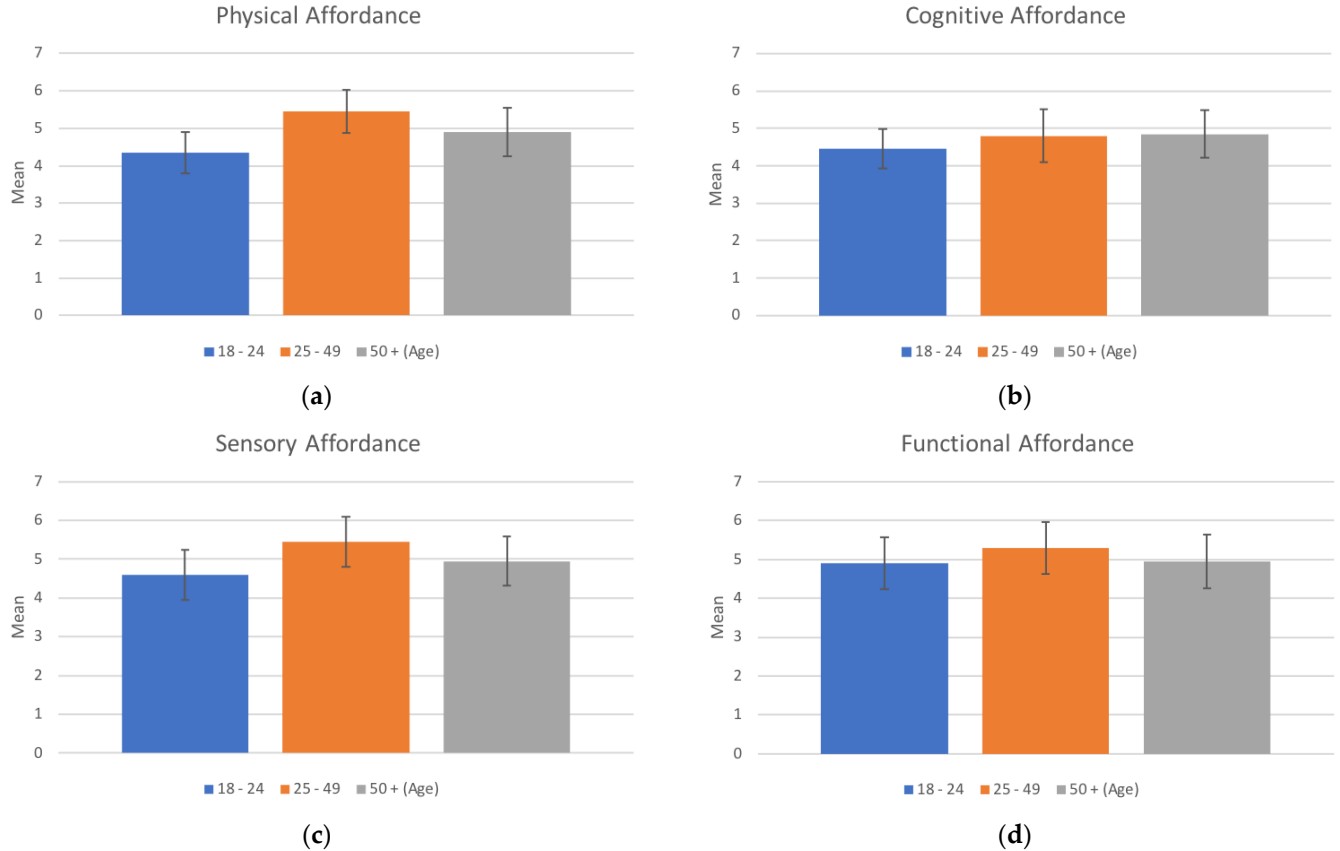

**Figure 3.** The mean affordances of smart speakers with the mechanical button control method for the three age groups. (**a**) Physical affordance; (**b**) cognitive affordance; (**c**) sensory affordance; (**d**) functional affordance. For each group, the error bar shows the 95% CI.

One-way ANOVA was used to analyze the independent variables, which are the survey questionnaires scores of the three age groups regarding the four affordances. There was no significant difference ($p > 0.01$) among the groups for all four affordances, as follows: for physical affordance, $F_{(2,57)} = 3.473$ and $p > 0.01$; for cognitive affordance, $F_{(2,57)} = 0.48$ and $p > 0.01$; for sensory affordance, $F_{(2,57)} = 1.783$ and $p > 0.01$; for functional affordance, $F_{(2,57)} = 0.42$ and $p > 0.01$ (Table 7).

**Table 7.** ANOVA results for smart speakers with the mechanical button control method.

| Affordance | Group | Square Root | df | Mean Square | F | Significance |
|---|---|---|---|---|---|---|
| Physical | Between groups | 12.100 | 2 | 6.050 | 3.473 | 0.038 |
| | Within group | 99.300 | 57 | 1.742 | | |
| | Total | 111.400 | 59 | | | |
| Cognitive | Between groups | 1.900 | 2 | 0.950 | 0.480 | 0.621 |
| | Within group | 112.700 | 57 | 1.977 | | |
| | Total | 114.600 | 59 | | | |
| Sensory | Between groups | 7.300 | 2 | 3.650 | 1.783 | 0.177 |
| | Within group | 116.700 | 57 | 2.047 | | |
| | Total | 124.000 | 59 | | | |
| Functional | Between groups | 1.900 | 2 | 0.950 | 0.420 | 0.659 |
| | Within group | 128.950 | 57 | 2.262 | | |
| | Total | 130.850 | 59 | | | |

### 3.2.2. No-Button–No-Touch Control Method

For the representative sample of the no-button–no-touch control method, a questionnaire with a seven-point Likert scale was used in the survey. Each participant answered all the questions. The mean score for physical affordance was 2.87, the score for cognitive affordance was 2.62, that for sensory affordance was 3.23, and that for functional affordance was 3.08. The mean of all four affordances was 2.95 (Table 8, Figure 4).

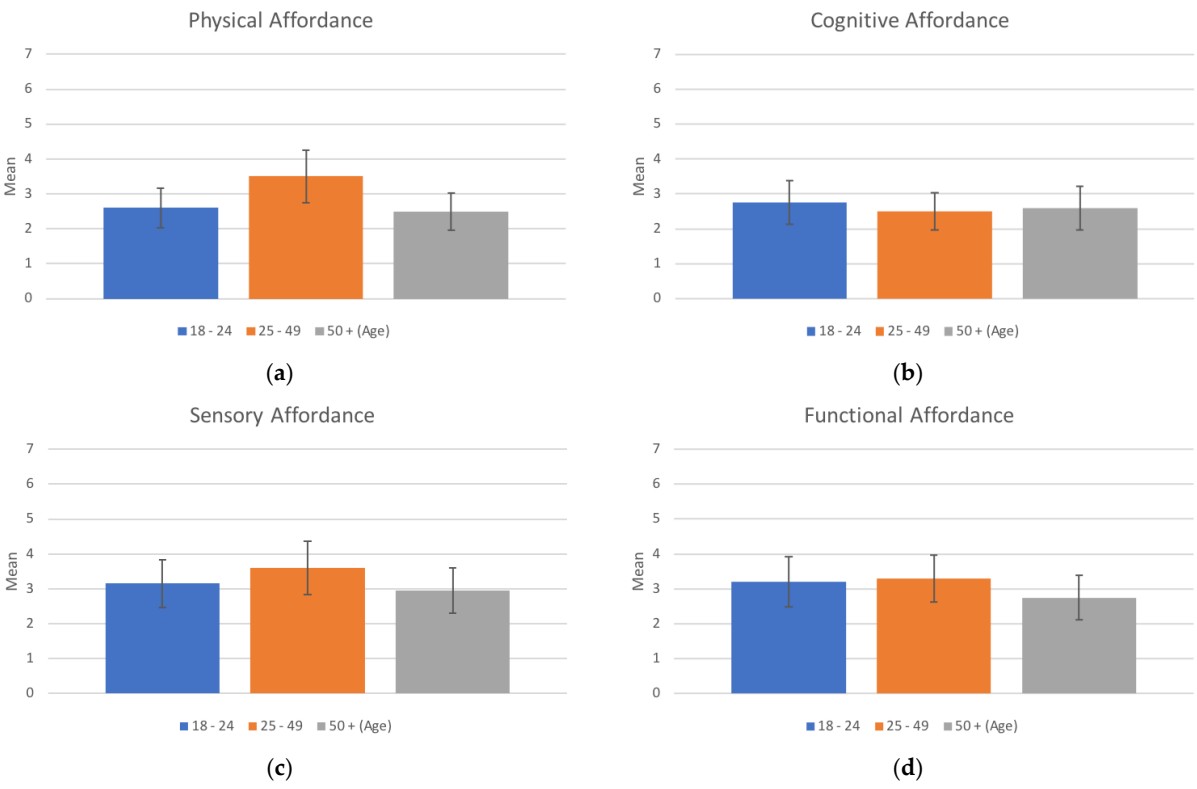

**Figure 4.** The mean affordances for smart speakers with the no-button–no-touch control method for the three age groups: (**a**) physical affordance; (**b**) cognitive affordance; (**c**) sensory affordance; (**d**) functional affordance. For each group, the error bar shows the 95% CI.

**Table 8.** Mean results for smart speakers with the no-button–no-touch control method.

| Affordance | Age Group | N | Mean | SD | SE |
|---|---|---|---|---|---|
| Physical | 18–24 | 20 | 2.60 | 1.273 | 0.285 |
| | 25–49 | 20 | 3.50 | 1.701 | 0.380 |
| | 50+ | 20 | 2.50 | 1.192 | 0.267 |
| | Total | 60 | 2.87 | 1.455 | 0.188 |
| Cognitive | 18–24 | 20 | 2.75 | 1.410 | 0.315 |
| | 25–49 | 20 | 2.50 | 1.192 | 0.267 |
| | 50+ | 20 | 2.60 | 1.392 | 0.311 |
| | Total | 60 | 2.62 | 1.316 | 0.170 |
| Sensory | 18–24 | 20 | 3.15 | 1.531 | 0.342 |
| | 25–49 | 20 | 3.60 | 1.698 | 0.380 |
| | 50+ | 20 | 2.95 | 1.468 | 0.328 |
| | Total | 60 | 3.23 | 1.566 | 0.202 |
| Functional | 18–24 | 20 | 3.20 | 1.609 | 0.360 |
| | 25–49 | 20 | 3.30 | 1.490 | 0.333 |
| | 50+ | 20 | 2.75 | 1.410 | 0.315 |
| | Total | 60 | 3.08 | 1.499 | 0.194 |

Note: N, number of participants; SD, standard deviation; SE, standard error.

One-way ANOVA was used to analyze the independent variables, which are the survey questionnaire scores of the three age groups for the four affordances. There was no significant difference ($p > 0.01$) among the groups for all four affordances. For physical affordance, $F_{(2,57)} = 3.066$ and $p > 0.01$; for cognitive affordance, $F_{(2,57)} = 0.178$ and $p > 0.01$; for sensory affordance, $F_{(2,57)} = 0.901$ and $p > 0.01$; and for functional affordance, $F_{(2,57)} = 0.758$ and $p > 0.01$ (Table 9).

**Table 9.** ANOVA results for smart speakers with the no-button–no-touch control method.

| Affordance | Group | Square Root | df | Mean Square | F | Significance |
|---|---|---|---|---|---|---|
| Physical | between groups | 12.133 | 2 | 6.067 | 3.066 | 0.054 |
| | Within group | 112.800 | 57 | 1.979 | | |
| | Total | 124.933 | 59 | | | |
| Cognitive | Between groups | 0.633 | 2 | 0.317 | 0.178 | 0.838 |
| | Within group | 101.550 | 57 | 1.782 | | |
| | Total | 102.183 | 59 | | | |
| Sensory | Between groups | 4.433 | 2 | 2.217 | 0.901 | 0.412 |
| | Within group | 140.300 | 57 | 2.461 | | |
| | Total | 144.733 | 59 | | | |
| Functional | Between groups | 3.433 | 2 | 1.717 | 0.758 | 0.473 |
| | Within group | 129.150 | 57 | 2.266 | | |
| | Total | 132.583 | 59 | | | |

### 3.2.3. Touchscreen Control Method

For the representative sample of the touchscreen control method, a seven-point Likert scale questionnaire was used in the survey, for which each participant answered all the questions. The mean score for physical affordance was 3.93, the score for cognitive affordance was 4.08, that for sensory affordance was 4.17, and that for functional affordance was 4.33. The mean for all four affordances was 4.13 (Table 10, Figure 5).

**Table 10.** Mean results for smart speakers with the touchscreen control method.

| Affordance | Age Group | N | Mean | SD | SE |
|---|---|---|---|---|---|
| Physical | 18–24 | 20 | 3.25 | 1.618 | 0.362 |
| | 25–49 | 20 | 5.15 | 1.04 | 0.233 |
| | 50+ | 20 | 3.4 | 1.603 | 0.358 |
| | Total | 60 | 3.93 | 1.666 | 0.215 |
| Cognitive | 18–24 | 20 | 3.2 | 1.824 | 0.408 |
| | 25–49 | 20 | 5.45 | 0.945 | 0.211 |
| | 50+ | 20 | 3.6 | 1.698 | 0.38 |
| | Total | 60 | 4.08 | 1.807 | 0.233 |
| Sensory | 18–24 | 20 | 3.35 | 1.785 | 0.399 |
| | 25–49 | 20 | 5.4 | 1.095 | 0.245 |
| | 50+ | 20 | 3.75 | 1.743 | 0.39 |
| | Total | 60 | 4.17 | 1.787 | 0.231 |
| Functional | 18–24 | 20 | 3.6 | 1.903 | 0.426 |
| | 25–49 | 20 | 5.35 | 0.813 | 0.182 |
| | 50+ | 20 | 4.05 | 1.638 | 0.366 |
| | Total | 60 | 4.33 | 1.674 | 0.216 |

Note: N, number of participants; SD, standard deviation; SE, standard error.

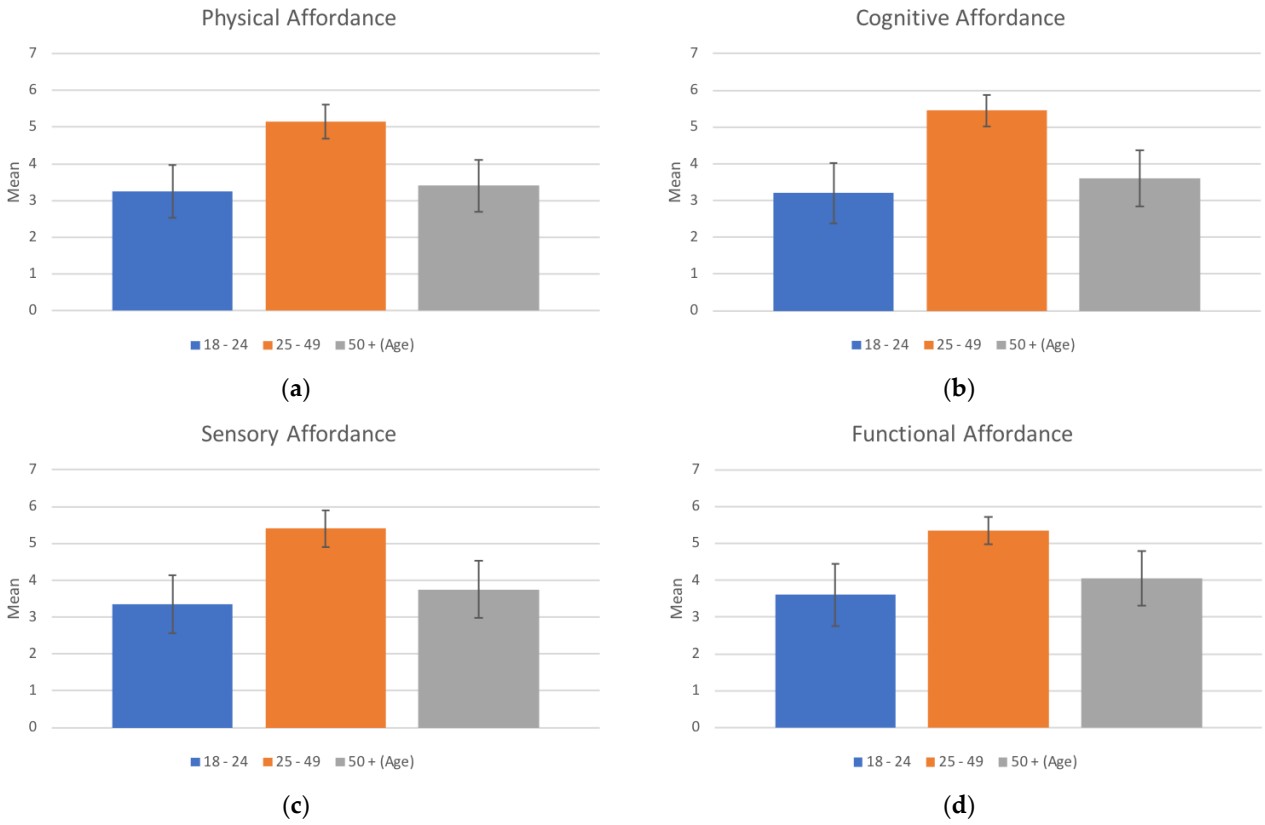

**Figure 5.** The mean affordances of speakers with the touchscreen control method for the three age groups. (**a**) Physical affordance; (**b**) cognitive affordance; (**c**) sensory affordance; (**d**) functional affordance. For each group, the error bar shows the 95% CI.

One-way ANOVA was used to analyze the independent variables, which are the survey questionnaire scores of the three age groups for the four affordances. There was significant difference ($p < 0.01$) among the groups for all four affordances, as follows: for physical affordance, $F_{(2,57)} = 10.681$ and $p < 0.01$; for cognitive affordance, $F_{(2,57)} = 12.172$ and $p < 0.01$; for sensory affordance, $F_{(2,57)} = 9.540$ and $p < 0.01$; for functional affordance, $F_{(2,57)} = 7.116$ and $p < 0.01$ (Table 11).

**Table 11.** ANOVA results for speakers with the touchscreen control method.

| Affordance | Group | Square Root | df | Mean Square | F | Significance |
|---|---|---|---|---|---|---|
| Physical | Between groups | 44.633 | 2 | 22.317 | 10.681 | 0.000 * |
|  | Within group | 119.100 | 57 | 2.089 |  |  |
|  | Total | 163.733 | 59 |  |  |  |
| Cognitive | Between groups | 57.633 | 2 | 28.817 | 12.172 | 0.000 * |
|  | Within group | 134.950 | 57 | 2.368 |  |  |
|  | Total | 192.583 | 59 |  |  |  |
| Sensory | Between groups | 47.233 | 2 | 23.617 | 9.540 | 0.000 * |
|  | Within group | 141.100 | 57 | 2.475 |  |  |
|  | Total | 188.333 | 59 |  |  |  |
| Functional | Between groups | 33.033 | 2 | 16.517 | 7.116 | 0.002 * |
|  | Within group | 132.300 | 57 | 2.321 |  |  |
|  | Total | 165.333 | 59 |  |  |  |

Note: $p < 0.01$ *.

### 3.2.4. Touch Sensor Control Method

For the representative sample of the touch sensor control method, a seven-point Likert scale questionnaire was used in the survey. Each participant answered all the questions. The mean score for physical affordance was 3.97, the score for cognitive affordance was 4.67, that for sensory affordance was 3.98, and that for functional affordance was 4.13. The mean for all four affordances was 4.19 (Table 12, Figure 6).

**Table 12.** Mean results for smart speakers with the touch sensor control method.

| Affordance | Age Group | N | Mean | SD | SE |
|---|---|---|---|---|---|
| Physical | 18–24 | 20 | 3.4 | 0.754 | 0.169 |
| | 25–49 | 20 | 5.15 | 1.348 | 0.302 |
| | 50+ | 20 | 3.35 | 1.04 | 0.233 |
| | Total | 60 | 3.97 | 1.353 | 0.175 |
| Cognitive | 18–24 | 20 | 4.3 | 1.261 | 0.282 |
| | 25–49 | 20 | 5.45 | 0.999 | 0.223 |
| | 50+ | 20 | 4.25 | 0.967 | 0.216 |
| | Total | 60 | 4.67 | 1.203 | 0.155 |
| Sensory | 18–24 | 20 | 3.35 | 1.268 | 0.284 |
| | 25–49 | 20 | 5.2 | 1.056 | 0.236 |
| | 50+ | 20 | 3.4 | 1.314 | 0.294 |
| | Total | 60 | 3.98 | 1.479 | 0.191 |
| Functional | 18–24 | 20 | 3.55 | 1.191 | 0.266 |
| | 25–49 | 20 | 5 | 1.124 | 0.251 |
| | 50 + | 20 | 3.85 | 1.182 | 0.264 |
| | total | 60 | 4.13 | 1.308 | 0.169 |

Note: N, number of participants; SD, standard deviation; SE, standard error.

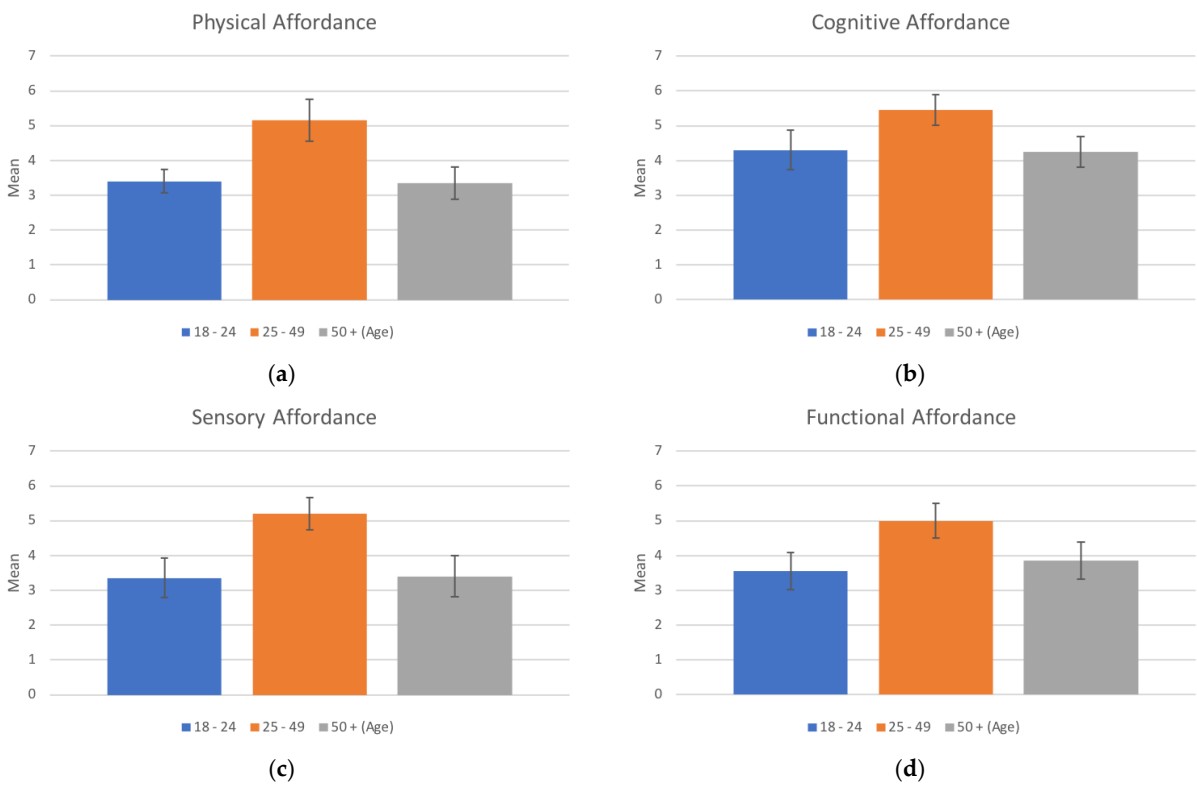

**Figure 6.** The mean affordances of smart speakers with the touch sensor control method for the three age groups. (**a**) Physical affordance; (**b**) cognitive affordance; (**c**) sensory affordance; (**d**) functional affordance. For each group, the error bar shows the 95% CI.

One-way ANOVA was used to analyze the independent variables, which are the survey questionnaires scores of the three age groups for the four affordances. There was a significant difference ($p < 0.01$) among the groups for all four affordances, as follows: for physical affordance, $F_{(2,57)} = 18.178$ and $p < 0.01$; for cognitive affordance, $F_{(2,57)} = 7.853$ and $p < 0.01$; for sensory affordance, $F_{(2,57)} = 14.978$ and $p < 0.01$; for functional affordance, $F_{(2,57)} = 8.617$ and $p < 0.01$ (Table 13).

**Table 13.** ANOVA results for smart speakers with the touch sensor control method.

| Affordance | Group | Square Root | df | Mean Square | F | Significance |
|---|---|---|---|---|---|---|
| Physical | Between groups | 42.033 | 2 | 21.017 | 18.178 | 0.000 * |
| | Within group | 65.900 | 57 | 1.156 | | |
| | Total | 107.933 | 59 | | | |
| Cognitive | Between groups | 18.433 | 2 | 9.217 | 7.853 | 0.001 * |
| | Within group | 66.900 | 57 | 1.174 | | |
| | Total | 85.333 | 59 | | | |
| Sensory | Between groups | 44.433 | 2 | 22.217 | 14.978 | 0.000 * |
| | Within group | 84.550 | 57 | 1.483 | | |
| | Total | 128.983 | 59 | | | |
| Functional | Between groups | 23.433 | 2 | 11.717 | 8.617 | 0.001 * |
| | Within group | 77.500 | 57 | 1.360 | | |
| | Total | 100.933 | 59 | | | |

Note: $p < 0.01$ *.

## 4. Discussion

Participants from the three different age groups (younger age, middle age, and older age) were tested regarding the four affordances (physical, cognitive, sensory, and functional) of four representative smart speaker samples, and this revealed different results.

Firstly, the mechanical button control smart speaker had the highest mean affordances among all the representative samples. This result shows that participants understand better how to control or use this product based on the affordance of this smart speaker. The ANOVA results showed that this sample has no statistically significant difference ($p > 0.01$) for all four affordances (physical, cognitive, sensory, functional) among the three age groups, which indicates that the null hypothesis cannot be rejected. For this smart speaker, participants might have more experience of using a general home audio system, or they may already know how to use it. The control method of a general home audio system normally applies mechanical buttons, and this smart speaker is also controlled by mechanical buttons. Therefore, participants might be more familiar with the appearance of the control method, and, thus, they gave the highest scores to this smart speaker. There are other home devices that use a similar control method in the market, such as televisions, CD players, and radios, and the users' experience with these products may have influenced this result regarding the affordance.

Secondly, the no-button–no-touch control method had the lowest mean affordances of all the representative samples. This result showed that the participants lack understanding about how to control or use this product based on the affordance of this type of smart speaker. The ANOVA results showed that this smart speaker sample had no statistically significant difference ($p > 0.01$) in all four affordances (physical, cognitive, sensory, functional) among the three age groups, which indicates that the null hypothesis cannot be rejected. The reasons indicate that the participants might have less experience with using a device with no-button–no-touch control method. Basically, the control method for this smart speaker is completely different from that of an ordinary home audio system. Therefore, participants may not be familiar with the no-button–no-touch control method for smart speakers. In the market, products with a similar control method are rare, so the users' lack of experience with similar products may have influenced this result regarding the affordance.

Thirdly, the touchscreen control method had a mean affordance in the middle range of all the representative samples. The mean for the middle age group was higher than that of the younger age and older age groups, which implies that the middle age group participants better understood the touch screen control method. The ANOVA results showed that this sample had a statistically significant difference ($p < 0.01$) for the four affordances among the three age groups, so the null hypothesis was rejected in favor of the alternative hypothesis. Smart speakers with touchscreen control have a similar appearance to a tablet or a larger smartphone, and the middle age group might have more experience with using this control method. This result indicates that different age groups might have different levels of knowledge and experience of products with the touchscreen control method. In the market, there are other devices that apply similar control methods, such as smartphones and tablets, and the users' experience with these products may have produced this result regarding the affordance.

Fourthly, the touch sensor control method had a mean affordance in the middle range of all the representative samples. The mean for the middle age group was higher than that of the younger age and older age groups, which implies that the middle age group participants better understood the touch sensor control method. The ANOVA results showed that this sample had a statistically significant difference ($p < 0.01$) for the four affordances among the three age groups, and, thus, the null hypothesis was rejected in favor of the alternative hypothesis. The appearance of a smart speaker with the touch sensor control method usually shows a control sign or symbol on the case cover, but the touch area is not as obvious as a mechanical button, which might affect the affordance. The middle age group might have more experience with using this control method, so they gave higher scores compared with the other age groups. Other devices that use a similar control method are on the market, such as the touch sensor lights or microwave touch sensors. The users' experience with these products may have produced this result regarding the affordance.

## 5. Conclusions

Norman (1999) stated that the appearance of a device could give important clues required for its appropriate operation [22]. This research shows there are different affordances among different age groups based on the appearance of a smart speaker's control method. Norman (1999) also explained that cultural constraints are the conventions shared by a cultural group, which can affect how a person operates a device [22]. This present study presents a review of the differences in the affordances of smart speakers among three age groups in Taiwan. Taiwan is in Asia, in which the culture and behavior are different from those of Western culture or other cultures in different countries. The results showed that the middle age group of Taiwanese better understood how to operate a smart speaker compared with other age groups, which might relate to most of them having jobs and a higher income. Those in that age range are usually financially self-sufficient, so they have a higher chance and more opportunity to buy and use more technological devices. On the other hand, the younger age group and older age group in Taiwan are mostly not financially self-sufficient, as they either need support from their parents or their children—many younger Taiwanese are students, and many older Taiwanese are retired or receive a lower income—and, thus, they have lower chance and less opportunity to buy and use more technological products, which would affect their familiarity and experience with using them. Therefore, this might be the cause of the different affordance results for smart speakers among the three age groups.

The age groups (18–24, 25–49, and over 50 years) had a significant difference in the four affordances (physical, cognitive, sensory, and functional affordance) for smart speakers with the touchscreen control method and the touch sensor control method. All participants were in the same cultural group, namely, local residents of Taiwan.

## 6. Future Recommendation

This research may contribute to new designs and redesigns of smart speakers in the future. A good smart speaker appearance design should allow people of different ages to easily understand how to control it based on the appearance of its control method, since the results show that the no-button-no-touch control, touchscreen control, and touch sensor control smart speakers do not have sufficiently clear affordances to different age groups. Researchers can redesign the appearance of these three representative smart speakers based on four affordances, and then conduct an experiment with the same criteria of this study, whose purpose is the verification and validation of the new designs, i.e., determining if they have better affordance results for all three age ranges. Furthermore, researchers can also investigate the affordances of smart speakers on different cultural groups in other countries, the results of which should be valuable.

**Author Contributions:** C.-F.W. supervised the entire study and designed the experiments. Y.-K.W. wrote the manuscript, provided conceptualization, and conducted the investigation and data analysis. H.-H.H. provided visualization and resources. C.-Y.H. carried out the investigation, conducted the experiment, and analyzed the data. All authors have read and agreed to the published version of the manuscript.

**Funding:** This research received no external funding.

**Institutional Review Board Statement:** The study was conducted in accordance with the Declaration of Helsinki, and approved by the Institutional Review Board (or Ethics Committee) of National Taiwan University (protocol code 202012ES066 and 15 March 2021).

**Informed Consent Statement:** Informed consent was obtained from all subjects involved in the study.

**Data Availability Statement:** Not applicable.

**Acknowledgments:** The authors would like to thank the research protocol of "A study on design specifications and human-machine interaction model of service robots for different task" from Tatung University, and thank the editor and reviewers for their constructive suggestions, which improved this manuscript.

**Conflicts of Interest:** The authors declare no conflict of interest.

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
