# Peer review of "Applying Affordance Factor Analysis for Smart Home Speakers in Different Age Groups: A Case Study Approach"

_sustainability, doi:10.3390/su14042156_

Round 1
Reviewer 1 Report
Dear authors,
The review report has been attached.

Author Response
Thank you very much for your advice and constructive suggestions, which improved this manuscript.
Please see the attachment.
Sincerely,
Ying-Kit Wong
Corresponding Author/ Second Author
Ph.D. Candidate
The Graduate Institute of Design Science,
Tatung University, Taipei 104, Taiwan
d10717004@ms.ttu.edu.tw

Reviewer 2 Report
The work is well structured. The authors present a review of smart speakers in Taiwan, focusing on the four main affordances (physical, cognitive, sensory, functional) and three different age groups (60 participants) based on four appearance categories of smart speaker control, namely mechanical button control, no-button– no-touch control, touchscreen control, and touch sensor control. As a suggestion to complete the work, I suggest the authors to detail the contribution of originality brought by the work.
Author Response

(The authors gave the same response as above.)

Reviewer 3 Report
The paper presents an affordance factor analysis for smart home speakers in different age groups. According to the reviewer’s opinion, the paper is well-structured and clear. The topic is interesting and falls within the aim of the journal. In addition, the results are well-presented and could be helpful to further develop the same topic. Therefore, the paper can be accepted for publication in the current form.
Author Response

(The authors gave the same response as above.)
